



# Small-scale variability of stratospheric ozone during the SSW 2018/2019 observed at Ny-Ålesund, Svalbard

Franziska Schranz[1], Jonas Hagen[1], Gunter Stober[1,2], Klemens Hocke[1,2], Axel Murk[1,2], and Niklaus Kämpfer[1,2]

[1]Institute of Applied Physics, University of Bern, Bern, Switzerland
[2]Oeschger Centre for Climate Change Research, University of Bern, Bern, Switzerland

**Correspondence:** Franziska Schranz (franziska.schranz@iap.unibe.ch)

**Abstract.** Middle atmospheric ozone, water vapour and zonal and meridional wind profiles have been measured with the two ground-based microwave radiometers GROMOS-C and MIAWARA-C. The instruments are located at the Arctic research base AWIPEV at Ny-Ålesund, Svalbard (79° N, 12° E) since September 2015. GROMOS-C measures ozone spectra in the four cardinal directions with an elevation angle of 22°. This means that the probed airmasses at an altitude of 3 hPa (37 km) have a

horizontal distance of 92 km to Ny-Ålesund. We retrieve four separate ozone profiles along the lines of sight and calculate daily mean horizontal ozone gradients which allow us to investigate the small-scale spatial variability of ozone above Ny-Ålesund. In winter 2018/2019 a major sudden stratospheric warming (SSW) took place with the central date at 2 January. We present the ozone, water vapour and wind measurements of the winter 2018/2019 and discuss the signatures of the SSW in a global context. We further present the evolution of the ozone gradients at Ny-Ålesund and link it to the planetary wave activity. At

3 hPa we find a distinct seasonal variation of the ozone gradients. The strong polar vortex during October and March results in a decreasing ozone volume mixing ration towards the pole. In November the amplitudes of the planetary waves grow until they break in the end of December and an SSW takes place. From November until February the ozone gradients mostly point to higher latitudes and the magnitude is smaller than in October and March. We attribute this to the planetary wave activity of wave number 1 and 2 which enabled meridional transport. The MERRA-2 reanalysis and the SD-WACCM model are able to

capture the small-scale ozone variability and its seasonal changes.

## 1  Introduction

In the Arctic, the polar vortex dominates the dynamics of the wintertime middle atmosphere. The polar vortex is a cyclonic wind system which forms in autumn from the balance between the Coriolis force and radiative forcing, which means the radiative cooling of the polar middle atmosphere in the absence of solar heating. The polar vortex maintains a transport barrier

between polar and midlatitude air which leads to gradients in trace gas concentrations across the polar vortex edge. Interactions of enhanced planetary waves with the mean flow can disturb this stable wind system (Matsuno, 1971) and cause a sudden stratospheric warming (SSW, Scherhag, 1952) which is the most dramatic meteorological phenomena in the middle atmosphere. Thereby the polar vortex can shift off the pole or even split into two or more sub vortices (Charlton and Polvani,





2007). The zonal mean wind reverses and in the stratosphere adiabatic descent leads to temperature increases up to 60 K or more within a few days whereas in the mesosphere adiabatic ascent leads to temperature decreases. Meridional transport and irreversible mixing across the polar vortex edge are enhanced during an SSW (Calisesi et al., 2001; Manney et al., 2009; Tao et al., 2015; de la Cámara et al., 2018).

Changes in dynamics and temperature during an SSW lead to drastic changes in the distribution of trace gases like ozone and water vapour. Ground-based microwave radiometry provides continuous profile measurements of these trace species and horizontal winds in the middle atmosphere with a high time resolution in the order of hours for the trace species and one day for wind. It is therefore a valuable technique for the investigation of the temporal changes in trace gas concentrations and dynamics on small timescales.

During an SSW the polar vortex often moves away from the pole to the midlatitudes. In the Arctic, this leads to the advection of midlatitude air to the pole where stratospheric ozone and mesospheric water vapour increases of up to 100 % and in the order of 50 % respectively were observed (Scheiben et al., 2012; Tschanz and Kämpfer, 2015; Ryan et al., 2016; Schranz et al., 2019). At Bern, Switzerland stratospheric ozone decreases of 30 % were measured during the major SSW in 2008. In the lower stratosphere the polar vortex passed at Bern and the decrease is explained with the advection of ozone poor polar

vortex air. In the upper stratosphere the polar vortex did not reach Bern and the ozone decreased mainly because increasing temperatures lead to faster ozone destruction via the $NO_x$ cycle (Flury et al., 2009). During the 2008 SSW mesospheric water vapour at Bern, Switzerland and Seoul, South Korea was anticorrelated. Whereas in Bern water vapour increased by 15 % (Flury et al., 2009), a water vapour decrease of 40 % was observed at Seoul because of altered transport patterns (De Wachter et al., 2011). The zonal wind reversals could be observed during several SSWs at midlatitudes and in the Arctic (Wang et al.,

2019; Rüfenacht et al., 2014; Schranz et al., 2019).

    The aforementioned studies observed a single profile per location and investigated the variability of trace species on small temporal scales. With the measurements from the ground-based microwave radiometer GROMOS-C we are for the first time able to investigate the variability of ozone on small spatial scales. GROMOS-C measures ozone spectra in the four cardinal directions under an elevation angle of 22°. From these spectra we retrieve four separate ozone profiles along the lines of sight

of GROMOS-C. This means that e.g. at an altitude of 37 km we observe ozone at four different locations which each have a horizontal distance to Ny-Ålesund of 92 km.

    Measurements of the spatial variability of trace gases on scales of a few hundred kilometres are rare. For ozone it was analysed by Sparling et al. (2006) in the upper troposphere and lower stratosphere to investigate the impact of small-scale variability on satellite data validation. They used high-resolution aircraft data and found that in general ozone varies about

4–12 % at 18–21 km in the lower stratosphere and about 15–25 % at 8–13 km in the upper troposphere across a scale of 150 km. Inside of the North and South polar vortices the variability is about 5 % whereas in the winter northern hemisphere (NH) outside of the polar vortex and poleward of 30° N the variability is 12–13% across the same scale. Anisotropy effects seem to be small on these scales even in the winter NH however flight paths at high latitudes were mostly across the polar vortex edge when it was distorted or off the pole, which could introduce a sampling bias as the authors note.



The ground-based microwave radiometers GROMOS-C and MIAWARA-C have been located at the Arctic research base AWIPEV at Ny-Ålesund, Svalbard (79° N/12° E) since September 2015 (Schranz et al., 2018, 2019). The instruments measure the thermal emission lines of ozone and water vapour, from which we retrieve middle-atmospheric volume mixing ratio (VMR) profiles and zonal and meridional wind profiles. In this paper we discuss the measurements in the context of the SSW which

took place in winter 2018/2019. Further we present the evolution of the small-scale ozone gradients above Ny-Ålesund during winter 2018/2019 and especially during the SSW and link it to the planetary wave activity.

The remainder of this article is organized as follows. Section 2 introduces the ground-based microwave radiometers and the model and reanalysis datasets used. A characteristics of the SSW 2018/2019 and the measurements from Ny-Ålesund are presented in Sect. 3. The small-scale spatial variability of ozone is discussed in Sect. 4. Summary and conclusion are given in

Sect. 5.

## 2 Instruments and models

In this article we used ozone, zonal and meridional wind and water vapour measurements from our two ground-based microwave radiometers GROMOS-C and MIAWARA-C. The instruments were both built at the University of Bern and are specifically designed for campaigns. This means that they are compact and operate autonomously with very little maintenance.

Since September 2015 the instruments have been located at the Arctic research base AWIPEV at Ny-Ålesund, Svalbard (79° N , 12° E) in the frame of a collaborative campaign of the University of Bremen and the University of Bern. Additionally we used temperature measurements from EOS-MLS on board of the Aura Satellite and ozone and wind data from the MERRA-2 reanalysis and ozone and water vapour from the SD-WACCM model.

### 2.1 GROMOS-C

GROMOS-C (GRound-based Ozone MOnitoring System for Campaigns) is a microwave radiometer which measures the pressure broadened rotational emission line of ozone at 110.8 GHz. GROMOS-C has an uncooled single-side-band heterodyne receiver system and a Fast Fourier Transform (FFT) spectrometer with 1 GHz bandwidth and 30.5 kHz spectral resolution. The system noise temperature of the instrument is about 1080 K. A detailed description of GROMOS-C is presented in Fernandez et al. (2015).

From the ozone spectra we retrieve 2 hourly ozone profiles which cover an altitude range of 23–70 km. We use the software QPACK (Eriksson et al., 2005) and ARTS2 (Eriksson et al., 2011) to perform the retrieval according to the optimal estimation method by Rodgers (1976). From the ozone spectra measured in the four cardinal directions we retrieve zonal and meridional wind profiles with the Doppler microwave radiometry method described in Hagen et al. (2018) and Rüfenacht et al. (2012). The retrieved wind profiles have a time resolution of 1 day and cover an altitude range from 75 km down to 60–45 km depending

on the tropospheric opacity.

Before the Ny-Ålesund campaign GROMOS-C was located at La Reunion (21° S) where a comparison with measurements from EOS-MLS showed an agreement within 5 % (Fernandez et al., 2016). At Ny-Ålesund Schranz et al. (2019) performed a



thorough intercomparison over 3 years with the satellite instruments EOS-MLS and ACE-FTS, with the model SD-WACCM and the reanalysis ERA5 and with OZORAM a ground based microwave radiometer also located at Ny-Ålesund (Palm et al., 2010) and with balloon borne ozone sonde measurements. On average the GROMOS-C measurements are within 6 % of the other datasets.

### 2.1.1 Measurement Geometry

The main purpose of GROMOS-C is to measure ozone spectra which allow the retrieval of ozone profiles in the middle atmosphere. From ozone spectra measured in opposite directions at a low elevation angle it is possible to retrieve a wind profile (Rüfenacht et al., 2012). Therefore GROMOS-C has a special observation system and ozone spectra are consecutively measured in all four cardinal directions with a repetition time of 4 s. The beam has a full width at half maximum of 5° and the measurements are performed under an elevation angle of 22°. This means that at an altitude of 37 km (3 hPa) the probed airmass is already 92 km away from the instrument location as shown in Fig. 1. The ozone profiles are retrieved separately in the four cardinal directions. With this dataset of continuous ozone measurements at four different locations we investigate small-scale ozone gradients in the middle atmosphere during the winter 2018/2019. We compare the results to ozone gradients from the MERRA-2 reanalysis and the SD-WACCM model, the locations of the model grid points are also indicated in Fig. 1.

### 2.1.2 GROMOS-C wind measurements

From the ozone spectra measured in the four cardinal directions we retrieve daily mean zonal and meridional wind profiles with the same method as described in (Hagen et al., 2018). Figure 2 and 3 show the time series of zonal and meridional wind speed retrieved from the GROMOS-C spectra. The grey horizontal lines indicate the upper and lower bound of a measurement response of 0.5 which we define as the trustworthy altitude range of the measurement. The measurement response is defined as the area below the averaging kernel of a given altitude and indicates the sensitivity of the retrieval (Rodgers, 2000). For wind retrievals the a priori profile is 0 m/s to allow positive and negative wind speeds with the same probability. This is especially important for the observation of sudden wind reversals in the context of extreme events. The grey background indicates data gaps or days where the retrieval did not converge because of too high noise levels when the opacity of the troposphere was high or when the measurement response was smaller than 0.5 for the whole profile.

Compared to the microwave radiometers (WIRA, Rüfenacht et al. (2012) and WIRA-C, Hagen et al. (2018)), which were specifically designed for wind measurements, GROMOS-C has a lower measurement response and the wind profiles cover a smaller altitude range. This is because they measure the ozone line at a higher frequency (142 GHz) where the Doppler shift is larger, the instrument noise temperature is lower and the used spectrometers have a higher spectral resolution.

The comparison of zonal and meridional wind measurements with convolved MERRA-2 data shows a good agreement (Fig. 2 and 3). Several wind reversals in the mesosphere from November–January are captured. Even in the stratosphere where the measurement response is below 0.5 the westward wind in the first days of January and the predominantly eastward winds in October and November are captured as well as the strong northward wind components before and after the SSW.





## 2.2 MIAWARA-C

MIAWARA-C (MIddle Atmospheric WAter vapour RAdiometer for Campaigns) is a ground-based microwave radiometer which measures the pressure broadened rotational emission line of water vapour at 22 GHz. The instrument has an uncooled heterodyne receiver system and an FFT spectrometer with 400 MHz bandwidth and a spectral resolution of 30.5 kHz. The system noise temperature of MIAWARA-C is about 150 K. From the measured spectra we retrieve water vapour profiles with QPACK (Eriksson et al., 2005) and ARTS2 (Eriksson et al., 2011), using an optimal estimation method (Rodgers, 1976). The profiles cover an altitude range of 37–75 km with a time resolution of 2–4 hours, depending on the opacity of the troposphere. A detailed description of the instrument and the retrieval algorithm is given in Straub et al. (2010) and Tschanz et al. (2013).

MIAWARA-C was located at Sodankylä and Bern in the years 2010–2013 and is located at Ny-Ålesund since September 2015. At Bern and Sodankylä an offset of +13 % compared to satellite measurements was seen in the mesosphere but in the upper stratosphere the measurements agreed mostly within ±5 % (Tschanz et al., 2013). A comparisons at Ny-Ålesund with EOS-MLS over 3 years shows an average offset over the full altitude range of 10–15 % depending on altitude but constant in time. The median relative difference of MIAWARA-C measurements to SD-WACCM simulations and measurements from the ACE-FTS satellite instrument is within ± 5 % on average (Schranz et al., 2019).

## 2.3 EOS-MLS

EOS-MLS is is the Earth Observing System Microwave Limb Sounder on board of NASA's Aura satellite (Waters et al., 2006). It was launched in 2004 into a sun synchronous orbit with 98° inclination and a period of 98.8 min. At Ny-Ålesund it passes twice a day at about 04:00 and 10:00 UT. We use the version 4.2 temperature product (Schwartz et al., 2015). The temperature profiles are derived from the 118 and 240 GHz radiometers and cover an altitude range from 10 to 90 km.

## 2.4 SD-WACCM

SD-WACCM (Brakebusch et al., 2013) is the specified dynamics version of NCAR's Whole Atmosphere Community Climate Model (WACCM, Marsh et al., 2013) and a component of the Community Earth System Model (CESM). The model grid extends from ground to 145 km altitude using 88 levels with a vertical resolution of 0.5–4 km. The spatial resolution is 1.9° latitude × 2.5° longitude and the temporal resolution is 30 minutes. In SD-WACCM the dynamics is constrained by meteorological analysis fields from GEOS5 (Rienecker et al., 2008). This means that at every model time step horizontal winds, temperature, surface wind stress, surface pressure and specific and latent heat flux are nudged towards the analysis fields in order to keep a realistic representation of the dynamics. The nudging strength is 10 % and it is performed up to an altitude of 70 km with a transition from 10 % to 0 % nudging between 70 km and 75 km. The chemistry module is based on MOZART the model for ozone and related chemical tracers (Emmons et al., 2010). The ozone variability in the Arctic middle atmosphere related to photochemical reactions is represented realistically (Schranz et al., 2018).



A previous comparison with GROMOS-C and MIAWARA-C at Ny-Ålesund over 3 years between 2015 and 2018 showed a median relative difference of 5 % up to 0.7 hPa for ozone and for water vapour it is within ±5 % up to 0.1 hPa (Schranz et al., 2019).

## 2.5    MERRA-2

The Modern-Era Retrospective Analysis for Research and Applications, version 2 (MERRA-2,  Gelaro et al., 2017) is the latest atmospheric reanalysis from NASA's Global Modeling and Assimilation Office (GMAO) (2015). It is calculated on a cubed-sphere grid with a resolution of $0.5° \times 0.625°$ and spans from the surface up to 0.01 hPa using 72 vertical levels. Measurements are assimilated in a 3D-Var assimilation scheme. Temperature and ozone profile measurements from EOS-MLS are used to also assimilate data in the upper stratosphere and mesosphere. EOS-MLS temperature profiles are assimilated above 5 hPa and ozone profiles at 215–0.02 hPa. For the period when primarily EOS-MLS data were assimilated in the reanalysis (2003–2012) a comparison with MIPAS measurements was performed. It shows that MERRA-2 underestimates ozone VMRs by up to 5 % compared to MIPAS during winter (DJF) in the Arctic stratosphere (100–1 hPa) (Wargan et al., 2017).

## 3    Characteristics of the SSW in 2018/2019

During the winter 2018/2019 a major sudden stratospheric warming (SSW) took place. It was first discussed by Rao et al. (2019) and they stated that the SSW is neither a typical displacement nor a typical split event. According to the zonal mean zonal wind data from MERRA-2 the central date of the SSW is on 2 January 2019. It is defined as the first day when the zonal mean zonal wind reverses from eastward to westward at 60 °N and 10 hPa (Charlton and Polvani, 2007). In this section we give an overview of the meteorological background situation and discuss the observations at Ny-Ålesund in a global context.

### 3.1    Meteorological background situation

An overview of the zonal mean temperatures measured with EOS-MLS and zonal mean zonal wind from MERRA-2 at 10 hPa in the stratosphere (Fig. 4) and at 0.1 hPa in the mesosphere (Fig. 5) for the winter 2018/2019 reveals the signatures of the SSW. At 10 hPa the latitudinal temperature gradient from 60–90° N reversed on 25 December and stayed reversed for one month. At 80° N the zonal mean temperature increased by about 45 K in less than a week. The reversal of the latitudinal temperature gradient at 10 hPa was accompanied by a reversal of the zonal mean zonal wind between 60°and 90° N which classifies the warming event as as a major sudden stratospheric warming (McInturff, 1978). The reversal of the wind also lasted for about 1 month. In the stratosphere the polar vortex reformed in February and stayed undisturbed until the end of March. In the mesosphere at 0.1 hPa the zonal mean temperature dropped about by 35 K at 80° N. The wind reversed for about a week at 60° N and for about 2.5 weeks at 80° N. Already in mid January the polar vortex reformed in the mesosphere and gained high wind speeds.

The evolution of the polar vortex during the SSW of the winter 2018/2019 is visualized in Fig. 6. The polar vortex edge is determined from ECMWF operational data as the geopotential height contour with the highest wind speed at a given pressure





level (Scheiben et al., 2012). The vortex started to shift notably around 20 December. In the mesosphere it was shifted towards Greenland whereas the stratospheric part was shifted towards Siberia. In the mesosphere the polar vortex started to be torn apart towards the end of December and split in 3 sub vortices on 31 December. In the stratosphere it eventually split on 3 January which is shortly after the central date of the SSW. At the same time in the mesosphere, the polar vortex already regained a

circular shape and the wind speed started to increase. On 12 January the polar vortex was reestablished in the mesosphere whereas in the stratosphere wind speeds are still very low and the algorithm detected the polar vortex edge at a latitude of 20° N which indicates a complete breakdown of the polar vortex system after the SSW in the stratosphere.

## 3.2 Observations at Ny-Ålesund

At Ny-Ålesund we were measuring ozone, water vapour and zonal and meridional wind profiles in the middle atmosphere
with our ground-based microwave radiometers. The SSW during the winter 2018/2019 was clearly visible in our data from Ny-Ålesund. Temperature measurements from EOS-MLS at Ny-Ålesund (Fig. 7, top) show that within one week stratospheric (10 hPa) temperatures increased by 50 K and the stratopause descended from about 0.15 hPa to 1.5 hPa. The temperature in the mesosphere (0.1 hPa) decreased by 50 K. After the SSW the stratopause was first indistinct and then reformed at a much higher altitude around 0.02 hPa. The stratopause height was then gradually decreasing until it reached about 1 hPa in the end
of March. The elevated stratopause is a phenomenon which often occurs after a split type SSW (Chandran et al., 2013) and has been previously observed (e.g. Manney et al., 2008). Matthias et al. (2012) compiled a composite of all SSW events between 1998 to 2011 using ECMWF data and MF-radar observations revealing the formation of an elevated stratopause in the mesospheric wind as strong zonal wind enhancement after the SSW. Later, Limpasuvan et al. (2016) made a composite analysis of SSWs with an elevated stratopause using WACCM data and explained the occurrence of an elevated stratopause
with strong, wave induced downwelling in the mesosphere which leads to adiabatic warming.

The zonal and meridional winds at Ny-Ålesund were retrieved from the GROMOS-C ozone spectra and compared to convolved and unconvolved MERRA-2 data (Figs. 2 and 3). The zonal wind was predominantly eastward in October and November. In the beginning of December it reversed to a westward direction. Except for a few days in mid December it stayed eastward for the whole month. At the time of the vortex split (Fig. 6d) we see strong westward winds in the stratosphere whereas in
the mesosphere the wind already reversed to an eastward direction. After the SSW a strong and stable vortex reforms in the mesosphere where higher wind speeds are measured than before the SSW which is in agreement with MF-radar observations at Andenes, Norway (69° N) (Matthias et al., 2012). The wind speeds in the stratosphere stay low until the polar vortex starts to reform in mid February.

The meridional wind speeds in autumn are low because the polar winter cyclone which dominates the dynamics in the Arctic
is centred at the pole. When the vortex shifted away from Ny-Ålesund towards Greenland and Canada in the end of December (Fig. 6b) very strong northward wind components were measured from the mesosphere down to the stratosphere. Shortly after the central date of the SSW, the stratospheric vortex split and meridional wind speeds were low. When the edge of the newly formed vortex was above Ny-Ålesund (Fig. 6e) this was leading to a very strong northward wind component for a second time.



This was followed by southward winds in mid January because the polar vortex was slightly shifted towards Siberia. From mid January on the meridional wind speeds are similar to the autumn period.

Figure 7 (middle) presents the ozone VMR time series measured with GROMOS-C in eastward direction. The ozone layer is clearly visible and the maximum of the ozone VMR is at about 3 hPa (37 km). During the SSW the ozone VMR increased in the
upper and middle stratosphere and reached up to 6.5 ppm. Except for a short ozone decrease in the upper stratosphere around 12 January the ozone VMR stays enhanced compared to November for the rest of the winter. In October and February/March a prominent diurnal cycle is present in the mesosphere. Signatures of wave activity are found during November and again in March with largest amplitudes around 2 hPa. With a bandpass filter (Hocke and Kämpfer, 2008; Hocke, 2009) we find peak-to-peak amplitudes of 0.8 ppm and periods of 3–4 days in November. In March we find peak-to-peak amplitudes of 1 ppm and
periods of 1.5–2.5 days. The wave activity in November is also seen in the water vapour time series.

The water vapour measurements from MIAWARA-C are shown in Fig. 7 (bottom). In autumn the water vapour is descending inside of the polar vortex. For the period of 15 September – 1 November 2018 the effective descent rate of water vapour calculated from the 5.5 ppm isopleth is 360 m/day. This is slightly lower than in the years 2015–2017 where the average effective descent rate was 433 m/day (Schranz et al., 2019). The fit is however very sensitive to the start and end dates and
a shift of the end date to 30 October leads to an effective descent rate of 398 m/day. During the SSW we observe a sudden increase of 2.5 ppm around 0.3 hPa whereas at the same time in the stratosphere at 3 hPa the water vapour VMR dropped by 1 ppm. After the SSW the mesospheric vortex reformed and we find again a vertical descent of the mesospheric water vapour.

### 3.3 $O_3$ and $H_2O$ measurements in a global context

To show the local ozone and water vapour measurements from our microwave radiometers in a global context we use northern
hemisphere ozone and water vapour data from the SD-WACCM model and indicate the contour of the polar vortex in Fig. 8 and 9.

For ozone we chose an altitude of 10 hPa which lies within the main ozone VMR layer in the stratosphere (Fig. 8). Prior to the SSW in November and December, Ny-Ålesund was inside of the polar vortex and the ozone VMR was about 2.5–4 ppm. After the central date when the polar vortex split and was shifted away from Ny-Ålesund, ozone VMR reached 6.5 ppm. At
the same time the Aleutian anticyclone is moving to the pole and the absence of sunlight leads to a low-ozone pocket inside of the anticyclonic wind system at about 10 hPa. This effect was observed by Manney et al. (1995) and explained by Morris et al. (1998) and Nair et al. (1998): the air inside of the Aleutian anticyclone is dynamically isolated at high latitudes long enough such that the ozone VMR decreases and approaches the local photochemical equilibrium. These low-ozone pockets were previously observed at Thule, Greenland with ground-based microwave radiometers and a correlation between the ozone
VMR and the solar exposure time of the air parcel within the last 10 days was found (Muscari et al., 2007). The polar vortex completely breaks down at this altitude after the split and the elongated Aleutian anticyclone passes over Ny-Ålesund where the ozone VMR drops to 4.7 ppm above 15 hPa for 5 days. Below 15 hPa, where ozone lifetimes are longer, no ozone decrease is seen. The ozone VMR then continuously decreases until it reaches a minimum of 3.6 ppm in mid of March. The ozone VMR of SD-WACCM in the polar plots (e.g. inside of the Aleutian high the SD-WACCM ozone VMRs are lower than for





GROMOS-C) differs from the GROMOS-C measurements because of the difference in the altitude resolution of the two data sets. In the upper stratospheric water vapour measurements we see that the overpass of the Aleutian anticyclone does not affect the water vapour VMR.

For water vapour we show the MIAWARA-C measurements in a global context at 0.3 hPa in the mesosphere in Fig. 9. Prior to the SSW Ny-Ålesund was inside of the polar vortex. Shortly before the central date of the SSW, the polar vortex moved away from Ny-Ålesund and the water vapour VMR increased by 2.5 ppm within 4 days. The vortex regains its circular shape soon and water vapour is descending, leading to a decrease in VMR at 0.3 hPa and to a growing water vapour gradient across the vortex edge. The minimum VMR is reached in mid of February and in March air masses are again rising inside of the summer anticyclone and VMRs increase. SD-WACCM captures the water vapour variation at 0.3 hPa nicely during the SSW. It however reaches the minimum VMR already in the beginning of February.

Since September 2015, when GROMOS-C and MIAWARA-C were installed at Ny-Ålesund, 3 major SSW took place in the northern hemisphere. The two major SSW of March 2016 and February 2018 were discussed in (Schranz et al., 2019). The evolution of ozone and water vapour above Ny-Ålesund during the SSW of the winter 2018/2019 and the stratospheric final warming of March 2016 were similar. Both SSW led to stratospheric ozone increases when the polar vortex moved away from Ny-Ålesund and an ozone decrease above 15 hPa when a low ozone pocket inside of the Aleutian anticyclone passed Ny-Ålesund. In February 2018 ozone VMRs doubled when the polar vortex split. Increases of mesospheric water vapour were observed during all three SSWs. A pronounced elevated stratopause only developed after the 2018/2019 SSW.

## 4 Small-scale spatial variability of $O_3$

GROMOS-C measures ozone spectra in the four cardinal directions under an observation angle of 22° and a time resolution of 2 h (see Sect. 2.1.1). From these spectra, four separate ozone profiles are retrieved along the lines of sight. At an altitude of 3 hPa (37 km) the distance between the E/W and N/S measurement locations is 184 km. We use the measurements at these four locations to calculate daily mean horizontal ozone gradients above Ny-Ålesund. For intercomparison we also used ozone data from the SD-WACCM model and the MERRA-2 reanalysis. Figure 1 shows measurement locations of GROMOS-C at 3 hPa and the grid points of SD-WACCM and MERRA-2. The ozone profiles of SD-WACCM and MERRA-2 were convolved with the averaging kernel of GROMOS-C before the gradients were calculated.

### 4.1 Ozone gradients at Ny-Ålesund

Figure 10 shows magnitude and angle of the daily mean horizontal ozone gradients above Ny-Ålesund over the course of the winter 2018/2019 and at an altitude of 3 hPa. An angle of $\varphi = 0°$ indicates an eastward pointing gradient, meaning that ozone VMRs are increasing towards the east. An angle of $\varphi = +90°$ indicates a northward pointing gradient. The gradients from the GROMOS-C data show a clear seasonal variation. In October ozone is increasing towards lower latitudes with on average 0.2 ppm/184 km. During November the gradients start to point to higher latitudes occasionally and in December they mainly reversed and indicate higher ozone VMR at higher latitudes. From December on the ozone gradients predominantly point





northwards until mid of February where the gradients suddenly turn southward and the magnitude increases and reaches up to 0.8 ppm/184 km. During the SSW the gradients first point in northeastern directions and then in northwestern directions. This is followed by strong eastward pointing gradients in the end of January. During the first half of February the gradients are again mostly pointing towards higher latitudes and are with a magnitude of 0.1 ppm/184 km smaller than the average magnitude of

0.2 ppm/184 km. The magnitude relative to the mean ozone VMR of the 4 cardinal directions is 4 %/184 km on average.

The general pattern of the ozone gradients at 3 hPa has already been seen in the 3 previous years of GROMOS-C observations at Ny-Ålesund. The mean over every day of the year of the north-south gradient from four years of GROMOS-C measurements shows that the gradients at 3 hPa are mainly pointing to lower latitudes from September until the beginning of November and then again during March. During summer and winter the gradients point mainly towards higher latitudes. This seasonal pattern

is observed in the upper stratosphere at altitudes between about 10 and 1 hPa.

Sparling et al. (2006) analysed the small-scale variability of ozone from measurements with an UV absorption instrument mounted on an aircraft. At 18–21 km altitude and inside of the NH polar vortex they found average relative differences of 4 % across a scale of 100 km. Differences in the magnitude of N/S and E/W gradients were not found at this altitude and at latitudes > 30° in winter. For GROMOS-C at 20 km altitude the distance between the N/S and E/W measurement locations is 100 km. In

the period where Ny-Ålesund was located inside of the polar vortex (October–March, except January) we find magnitudes of 0.1 ppm/100km on average which corresponds to a relative difference of 4 % and is in exact agreement with the measurements of Sparling et al. (2006).

## 4.2  Comparison with SD-WACCM and MERRA-2

We compared the evolution of the horizontal ozone gradients at 3 hPa, measured by GROMOS-C with MERRA-2 and SD-

WACCM (Fig. 10). The comparison shows that the reanalysis and the model capture the prominent features of the magnitude time series for example the low magnitudes in the beginning of February and the subsequent variability in March. On average the magnitude of the SD-WACCM gradients is 28 % lower and MERRA-2 is 11 % higher than GROMOS-C. The correlation coefficient with GROMOS-C is 0.7 for both SD-WACCM and MERRA-2. We use the Pearson's correlation coefficient which is defined as the covariance of the two datasets divided by the product of the standard deviations of the two datasets: $\rho =$

$\mathrm{cov}(A, B)/(\sigma_A \sigma_B)$. In October MERRA-2 shows about 0.11 ppm/184 km (about 60 %) higher magnitudes while the angles are still captured well. The angles agree again well in the end of January when the gradients were eastward pointing and then again in the end of February and during March. From November until the end of February the angles are less stable but in December and January they are mainly >0 for GROMOS-C whereas SD-WACCM and MERRA-2 also show negative angles. During October and March the angles of SD-WACCM and MERRA-2 are on average smaller than the angles of GROMOS-C

by 22°and 10° respectively.

## 4.3  Influence of planetary waves on local ozone gradients

The horizontal ozone gradients measured by GROMOS-C show strong variations throughout the observation period. Eixmann et al. (2019) investigated the temporal variability of the stratopause region temperatures using nightly averaged lidar and reanal-





ysis data and found that the day-to-day variability was mostly driven by stationary planetary waves 1,2 and 3. To characterize the influence of the planetary waves on the variability of the ozone gradients we calculated amplitude and phase of the stationary planetary waves in the MERRA-2 zonal and meridional wind fields using a wave diagnostics algorithm as described in Baumgarten and Stober (2019). We first present the evolution of the stationary planetary waves and discuss the dominant

gradient patterns of October, December and March in the context of the planetary wave activity. We further demonstrate the influence of the stationary planetary waves 1 and 2 on the ozone gradients by means of reconstructing local wind fields from the planetary waves.

Figure 11 shows amplitude and phase of the stationary planetary wave 1 and 2 in the zonal wind field of MERRA-2 at 79° N and 3 hPa. Waves with higher wave numbers have low amplitudes and are not considered here. The amplitudes of wave 1 and

2 start to increase in November and reach 60 m/s and 25 m/s respectively with a stable phase. The amplitude of wave 2 already decreases in mid December whereas the wave 1 is stable during December and shows a period of about 10 days. In the end of December the wave 1 breaks down and the SSW takes place. After the SSW the wave amplitudes stay below 20 m/s until mid of March.

Figure 12 shows SD-WACCM ozone at 3 hPa in the northern hemisphere, illustrating the different gradient patterns observed

in October, December and March at Ny-Ålesund. Additionally, the zonal and meridional wind components from SD-WACCM are shown for the same dates. In October the amplitudes of the planetary waves 1 and 2 are low and the polar vortex is mainly centred at the pole and the zonal wind field is zonally symmetric which leads to a transport barrier between midlatitude and polar air. At the latitude of Ny-Ålesund and at 3 hPa there is no net chemical ozone production during the winter season (October–mid March) which leads to a stable southward pointing ozone gradient. The net chemical ozone production is the

difference between the chemical ozone production and loss rate from SD-WACCM.

In November the amplitudes of wave 1 and 2 start to increase and the polar vortex is shifted towards Asia or Europe which leads to enhanced meridional transport and reverses the direction of the ozone gradients at Ny-Ålesund. The increasing wave activity reduces the latitudinal mixing barrier and the average amplitude of the ozone gradients decreases compared to October. The passing of filamentary ozone structures inside of the polar vortex leads to spikes in the gradient amplitudes. In February

after the SSW the planetary wave amplitudes are low, the polar vortex is again centred at the pole and the magnitude of the ozone gradient increases. In March the polar vortex is centred at the pole but does not have a circular shape as in October which leads to southward gradients with a large variability of the magnitudes.

To demonstrate the influence of the planetary waves 1 and 2 on the ozone gradients at Ny-Ålesund we reconstructed the zonal and meridional wind fields for Ny-Ålesund from the amplitude and phase of the planetary waves according to

$$u, v(t) = u_0, v_0(t) + \sum_{s=1}^{2} A_{u_s, v_s}(t) \cdot \cos(\lambda \cdot s - \phi_{u_s, v_s}(t)), \tag{1}$$

where $u_0$ and $v_0$ are the zonal mean zonal and meridional wind for the latitude of Svalbard, $A_{u_s, v_s}$ and $\phi_{u_s, v_s}$ are zonal and meridional amplitude and phase of the stationary planetary wave with wave number $s$ and $\lambda$ is the longitude of Ny-Ålesund. The reconstructed wind fields contain only information about the background wind and the waves 1 and 2, and therefore allow us to check if these components alone can explain the observed ozone gradients.





Figure 13 shows the angle of the reconstructed wind vector and the ozone gradient during winter 2018/2019. The correlation coefficient between the two angles time series is 0.4. The angle plot illustrates how the planetary wave activity and the ozone gradients are connected. At the beginning of the winter season in October and November a stable polar vortex evolves with a strong zonal wind, which blocks the meridional transport of ozone from the mid-latitudes into the polar cap resulting in a

90° angle difference between the ozone gradient (pointing southward) and the wind vector (eastward). During November the stationary planetary wave 1 and 2 start to grow in amplitude leading to a more disturbed polar vortex, which is reflected as the angle difference between the ozone gradient and the wind vector more or less disappeared towards the end of November. The planetary waves disrupt the blockage of the polar vortex and allow either that ozone rich air is transported into the polar cap or ozone poor air is advected to the mid-latitudes. Both processes affect the observed ozone gradients at Ny-Ålesund. This

ozone mixing due to the planetary waves occurs more or less during the whole winter season from December to February. The SSW event lasts not long enough that the westward wind regime in the stratosphere could again establish a stable blocking of the meridional transport of ozone. However, the angle shows during the SSW again a 90° angle offset relative to the ozone gradients, but opposite in sign. After the SSW the polar vortex remains rather weak and, although, the planetary wave 1 and 2 do not reach the same strength as before the SSW, they sustain the mixing of air between the mid- and polar latitudes. At the end

of February and in March the polar vortex recovers and establishes again the blockage of the meridional ozone transport due to strong eastward zonal winds, which is described by the 90° angle difference between wind vector and the ozone gradient. Considering the SD-WACCM results shown in Figure 12, it is obvious that the meridional transport of ozone is massively affected by the polar vortex and its distortion due to stationary planetary waves 1 and 2.

## 5  Conclusions

We presented the co-located observations of middle atmospheric ozone, water vapour and zonal and meridional wind profiles during the Arctic winter 2018/2019. The profiles were measured with two ground-based microwave radiometers which are located at the AWIPEV research base at Ny-Ålesund, Svalbard (79° N, 12° E). The ability to retrieve zonal and meridional wind profiles requires the measurement of ozone spectra in the four cardinal directions with a low observation angle. From these spectra four separate ozone profiles along the lines of sight are retrieved which gives a unique dataset to investigate

small-scale spatial ozone variability. At an altitude of 37 km (3 hPa) the probed air masses in E/W and N/S directions have a horizontal distance of 184 km. At Ny-Ålesund which is located at 79° N this distance between the East and West measurement corresponds to 9.5° longitude.

In winter 2018/2019 a major SSW took place. The central date, according to the MERRA-2 zonal wind reversal at 10 hPa, was on 2 January. At Ny-Ålesund temperatures were increasing 50 K in the stratosphere at 10 hPa and decreasing by the same

amount in the mesosphere at 0.1 hPa. The measured zonal and meridional wind speeds and ozone and water vapour VMR are highly depending on the location of the polar vortex during the SSW. At 10 hPa the ozone VMR almost doubled to 6.5 ppm when the polar vortex split. From SD-WACCM simulations we know that the net chemical ozone production is negative during the time period considered here. Therefore we can attribute the ozone increase to enhanced meridional transport. The passing of





the elongated Aleutian high pressure system containing a low ozone pocket reduced ozone VMR at Ny-Ålesund by 25 % for a few days. At 0.3 hPa we found strong increases in water vapour VMR of 50 % which were followed by a steady decrease when the polar vortex reformed and the air masses in the polar region were descending again. The wind field was highly variable and depends on the location of the polar vortex. The split of the polar vortex was visible in the wind measurements as enhanced

meridional wind speeds.

From the ozone measurements in the four cardinal directions we calculated daily mean local ozone gradients for the winter 2018/2019. At 20 km altitude (44 hPa) we find a relative magnitude of the ozone gradients of 4 % across a scale of 100 km and when Ny-Ålesund was inside of the polar vortex. This is in agreement with observations of Sparling et al. (2006). At higher altitudes we find a seasonal variation in the amplitude and the orientation of the ozone gradients. Strong local gradients in

southward direction occur in October and again in March when the polar vortex is stable and the planetary wave activity is low. From November on the amplitudes of the planetary waves 1 and 2 grows until they break down in the end of December and an SSW takes place. The ozone gradients mainly point northward during this time period and the magnitudes are lower. The MERRA-2 reanalysis and the SD-WACCM model capture the seasonal variation in magnitude and angle of the ozone gradients. To link the changes in the ozone gradients with the planetary wave activity we reconstructed the wind field at Ny-

Ålesund from amplitude and phase of the planetary waves 1 and 2. We found a correlation of 0.4 between the angle of the ozone gradients and the direction of the reconstructed wind. Our results indicate that the ozone mixing above Ny-Ålesund during the winter season 2018/2019 was driven by the planetary wave activity of wave number 1 and 2, which disturbed the polar vortex and enabled a meridional transport in- and outbound of the polar cap region reducing the observed spatial ozone gradients from GROMOS-C. The presented measurements of GROMOS-C and MIAWARA-C point out that the VMR ratio

of ozone and water vapour, is not only driven by the chemistry in polar stratospheric clouds, it is also affected by dynamical processes due to planetary waves.

*Code and data availability.* Ozone and water vapour measurements from the ground-based microwave radiometers GROMOS-C and MIAWARA-C are available at the NDACC data repository ftp://ftp.cpc.ncep.noaa.gov/ndacc/station/nyalsund/hdf/mwave/ and EOS-MLS temperature data are available at https://disc.gsfc.nasa.gov/datasets/ML2T_004/summary. MERRA-2 data were downloaded from

https://disc.gsfc.nasa.gov/datasets/M2I6NPANA_5.12.4/summary. The source code of CESM1.2.2 is available at http://www.cesm.ucar.edu/models/cesm1.2/.

*Author contributions.* FS was responsible for the ground-based ozone and water vapour measurements with GROMOS-C and MIAWARA-C, performed the data analysis and prepared the manuscript. JH performed the wind retrieval from the GROMOS-C ozone spectra. GS provided the wave diagnostics algorithm and contributed to the interpretation of the results. AM was responsible for the instrument development. FS,

NK and KH designed the concept of the study. All co-authors contributed to the manuscript preparation.



*Competing interests.* The authors declare that they have no conflict of interest.

*Acknowledgements.* Observations with MIAWARA-C and GROMOS-C in Ny-Ålesund are funded by the Swiss National Science Foundation and the German Research Foundation (DFG, Arctic Amplification: Climate Relevant Atmospheric and Surface Processes, and Feedback Mechanisms (AC)3 in projects B06 and E02). We thank the Jet Propulsion Laboratory/NASA for providing the EOS-MLS retrieval product,
5  the European Centre for Medium Range Weather forecasts (ECMWF) for providing operational analysis data, the Global Modeling and Assimilation Office (GMAO) at NASA Goddard Space Flight Center for providing MERRA-2 data and NCAR for providing the CESM/SD-WACCM source code. The CESM project is supported primarily by the National Science Foundation. We thank Mathias Palm from the University of Bremen, the electronics workshop of the IAP and the AWIPEV teams for their support during the campaign.



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



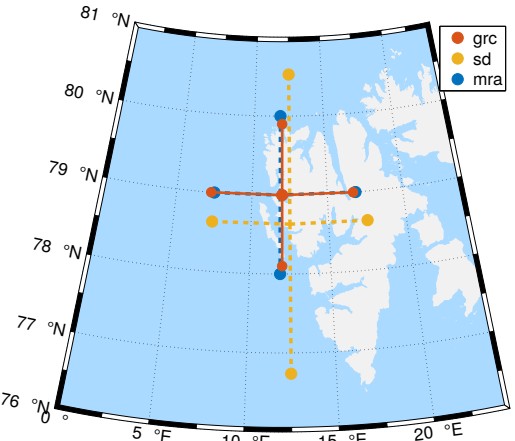

**Figure 1.** Location of the data points of GROMOS-C, MERRA-2 and SD-WACCM at 3 hPa (37 km) which are used to calculate the ozone gradient above Ny-Ålesund (indicated with a red square).

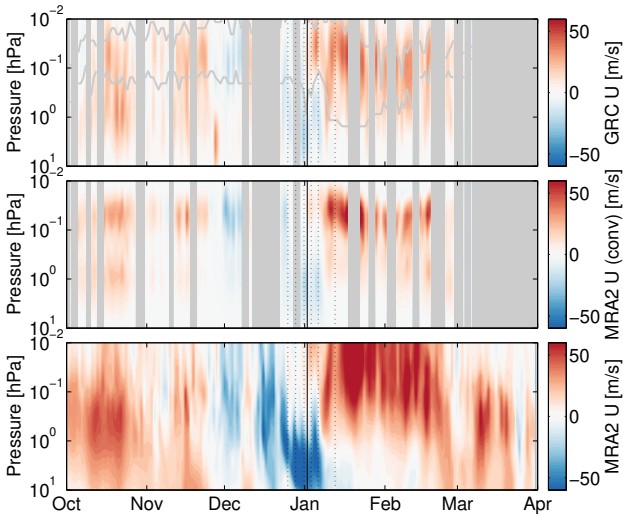

**Figure 2.** Zonal wind at Ny-Ålesund measured with GROMOS-C (top) and from the MERRA-2 reanalysis convolved with the averaging kernels of GROMOS-C (middle) and unconvolved (bottom) for the winter 2018/2019. The grey background indicates data gaps and the grey lines indicate the area where the measurement response is larger than 0.5. The dotted lines indicate the dates of the polar vortex snapshots in Fig. 6 and the solid line shows the central date of the SSW





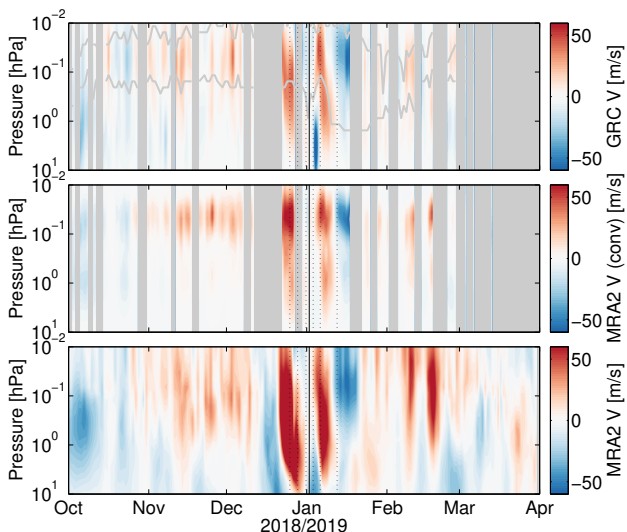

**Figure 3.** Same as in Fig. 2 but for meridional wind.

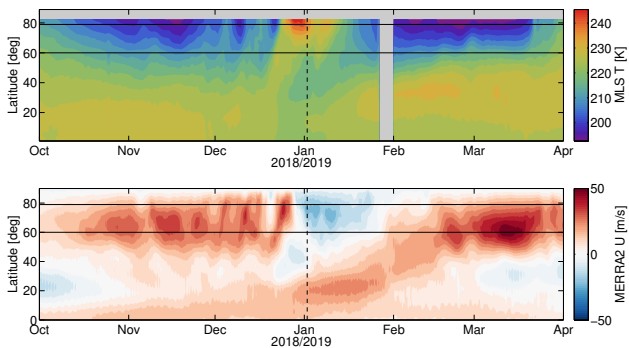

**Figure 4.** Zonal mean temperature from MLS and zonal mean zonal wind from MERRA-2 at 10 hPa in the stratosphere for the winter 2018/2019. The dashed line indicates the central date of the SSW. The horizontal lines indicate 60° N and 79° N.



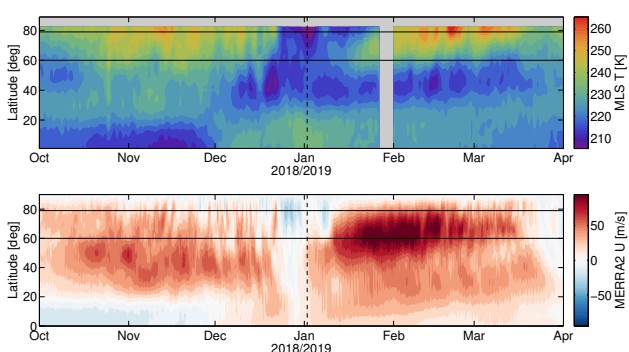

**Figure 5.** Same as in Fig. 4 but at 0.1 hPa in the mesosphere.

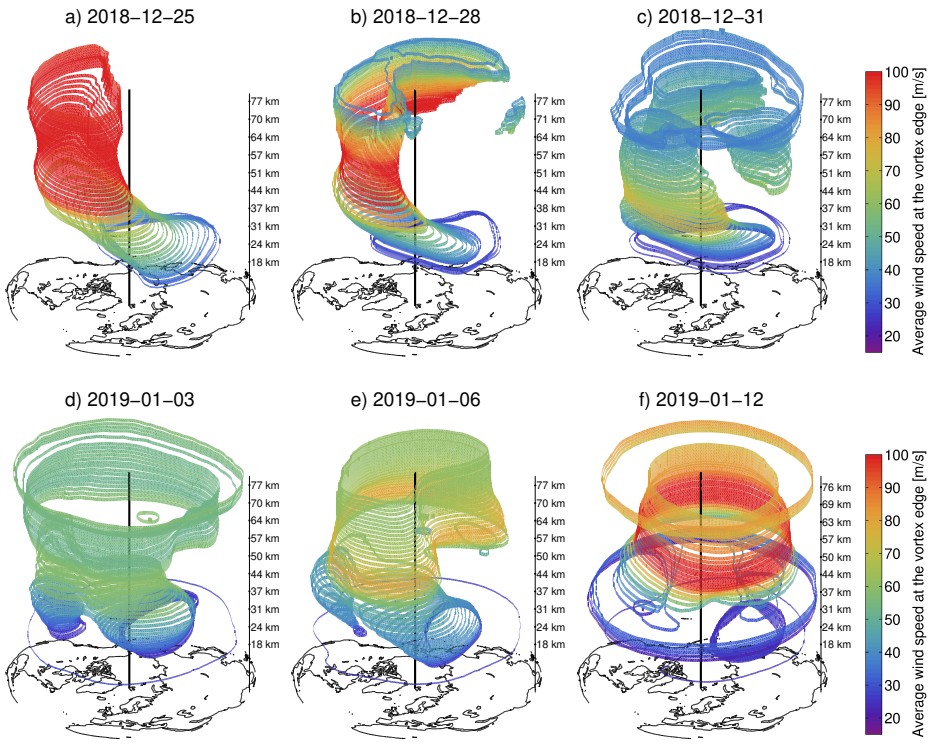

**Figure 6.** Contours of the polar vortex during the SSW in winter 2018/2019. The vertical black line is positioned at Ny-Ålesund.





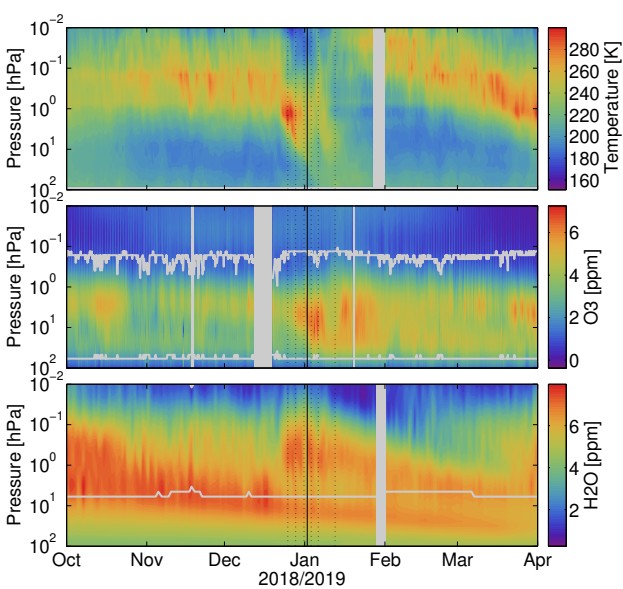

**Figure 7.** EOS-MLS temperature (top), GROMOS-C ozone VMR (middle) and MIAWARA-C water vapour VMR (bottom) time series at Ny-Ålesund for the winter 2018/2019. The dotted lines indicate the dates of the polar vortex snapshots in Fig. 6 and the solid line shows the central date of the SSW. The grey background indicates data gaps and the grey lines indicate the area where the measurement response is larger than 0.8 for GROMOS-C and MIAWARA-C.

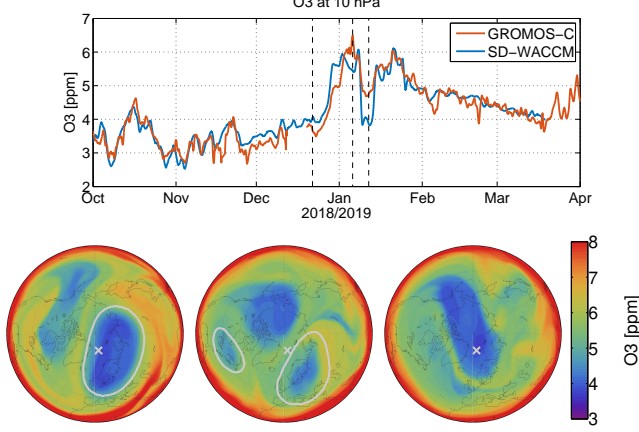

**Figure 8.** GROMOS-C and SD-WACCM ozone VMR time series at 10 hPa smoothed by a 1-day running mean (top) and SD-WACCM ozone VMR at 10 hPa in the northern hemisphere (bottom) for three dates which are indicated with dashed lines in the top panel. In the NH plots (bottom) the location of Ny-Ålesund and the polar vortex edge are indicated in grey.





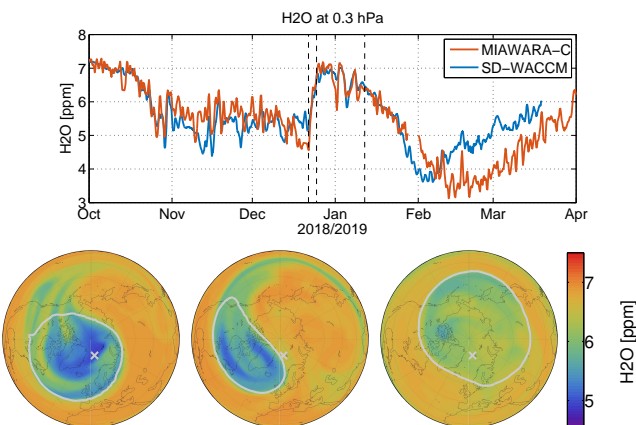

**Figure 9.** Same as in Fig. 8 but for MIAWARA-C water vapour VMR time series at 0.3 hPa.

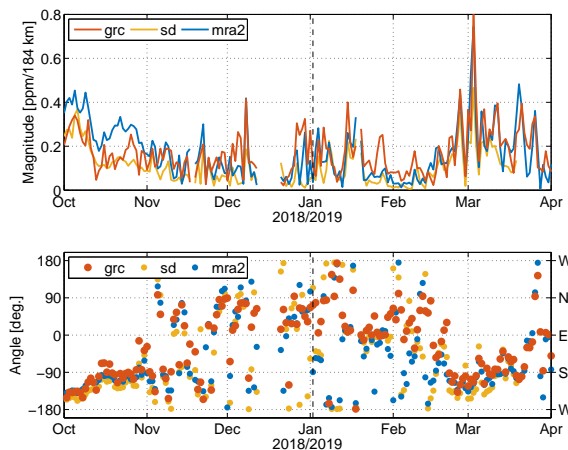

**Figure 10.** Magnitude and angle of the ozone gradients at 3 hPa above Ny-Ålesund from GROMOS-C, SD-WACCM and MERRA-2 data during the winter 2018/2019. The dashed line is at the central date of the SSW. The cardinal directions which correspond to an angle are indicated on the right.





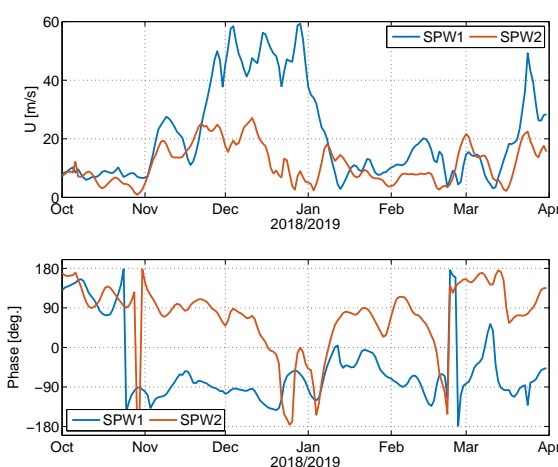

**Figure 11.** Amplitude (top) and phase (bottom) of the stationary planetary waves 1 and 2 calculated from the MERRA-2 zonal wind.

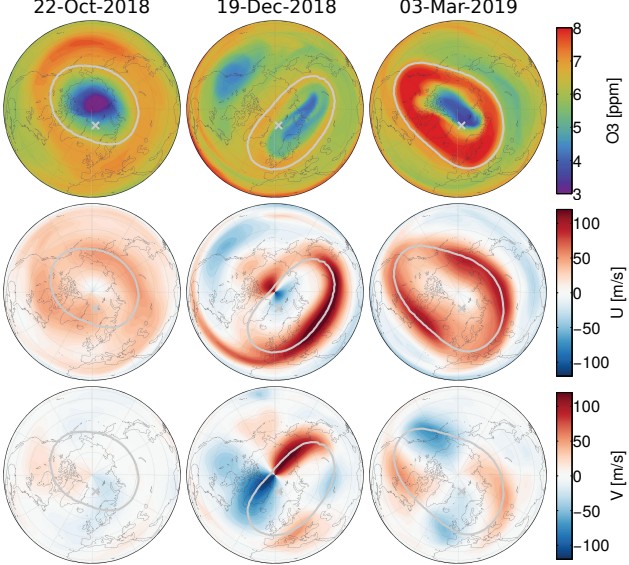

**Figure 12.** SD-WACCM ozone VMR, zonal wind and meridional wind at 3 hPa. The edge of the polar vortex and the location of Ny-Ålesund are indicated in grey.





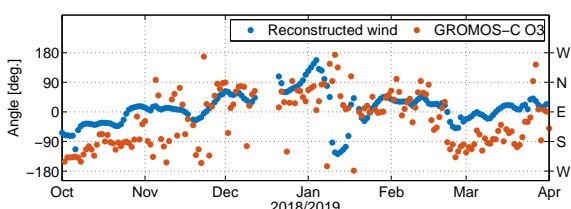

**Figure 13.** Direction of the horizontal wind field at Ny-Ålesund reconstructed from the stationary planetary waves 1 and 2 and the background wind and the angle of the GROMOS-C ozone gradients at 3 hPa.