# Peer review of "Small-scale variability of stratospheric ozone during the SSW 2018/2019 observed at Ny-Ålesund, Svalbard"

_Atmospheric Chemistry and Physics, 2019_

## Referee Comment (RC1) · Anonymous Referee #1 · 15 Jan 2020

This manuscript presents an analysis of the effects of the sudden stratospheric warming (SSW) of January 2019 on the evolution of stratospheric ozone observed from the Arctic research base at Ny-Alesund, Svalbard. The authors also analyze in detail the influence of wave activity of very long waves on ozone gradients. To do so, different types of data are used: observations from two ground-based microwave radiometers, atmospheric reanalysis (MERRA-2) and the output of specified-dynamics simulations of WACCM for the winter 2018-19. The results show that the SSW did have an effect on the stratospheric ozone distribution at Ny-Alesund. Although poleward transport is enhanced during the SSW and so, an ozone increase would be expected at high latitudes, the authors show that the ozone mixing ratio is highly dependent on the stratospheric

circulation, i.e. location of the Aleutian high and the polar vortex. Moreover, the authors find a link between an enhancement of wavenumber- 1 and 2 wave activity and a reduction in ozone gradients close to Ny-Alesund during the 2018/19 winter.

I find the analysis interesting, in particular, the study of the evolution of the ozone gradient in relation with the wave activity. However, I have some concerns that the authors should address before its publication. First of all, I would really encourage the authors to highlight more strongly the novel results of the study. In this sense, I think that the analysis of the small spatial scale variability of the ozone should have a more prominent place in the manuscript to make a stronger difference from the previous paper by Schranz et al. (2019). Secondly, the authors often refer to results that do not appear in the manuscript. In some cases, you can just say "(not shown)", but in other cases, I think it would be great to show the results. For instance, in P7L1, it is indicated that vortex started to shift notably around 20 December, but the vortex structure is only shown from 25 December. I think it would be great to show its structure before 20 December to appreciate the shift.

Other comments (line by line): P4 L18: Please specify that the grey horizontal lines are in Figures 2a and 3a. P6 L23-25: I guess you mean that "between 60° and 90°N" refers to the reversal of the latitudinal temperature gradient at 10hPa but not to the reversal of the zonal mean zonal wind. Typically, it is the reversal of the zonal wind at 60°N what defines the occurrence of a major SSW, as it is indicated some lines above in the same page. P6 L25: Please remove one "as" P6L30-31: Please describe more in detail the criterion for the identification of the polar vortex edge. P7 L23-24: I am confused. I am not sure which level the authors are referring to, because in the stratosphere and most of the mesosphere, the winds remain westward the whole December except for some days in mid-December. Please rewrite this sentence. P7 L29-30: Please add not shown. P8 L12: Figure 7 only shows the water vapor from 1st October. However, since the effective descent rate of $H_2O$ is measured for the period of 15 September-1 November, I would start the figure on 15 September too. P11 L14:

[Figure]

Northern Hemisphere P11 L15: Figure 12 shows only the plots for a specific day of October, December and March but not the monthly field. I recommend the authors computing the monthly means of the fields if they want to show the monthly state. P11 L21-22: I guess the authors are comparing figure 10 and 11, so please indicate it. P11 L24-27: The wave activity is already low in January and the ozone gradient is then increasing by that time. However, by the end of January and the first fortnight of February the ozone gradient decreases and the wave activity is still low. How do you explain that? P11 L33: Background wind –> zonal mean wind. P12 L 4: I think the wind is mainly southward in October. P12 L9-10: so, no specific effects of the SSW? Figure 1: I cannot see the red square indicating the location of Ny-Alesund.

---

## Referee Comment (RC2) · Anonymous Referee #2 · 19 Jan 2020

General comments: This study used the reanalysis and microwave radiometer measurements to investigate the January 2019 SSW event and its impact on the local stratospheric ozone distribution at Ny-Ålesund, Svalbard. The observational data are very useful in verifying this SSW event as reported in recent studies (Rao et al. 2019, 2020), and this manuscript is suitable for the main scope of the journal, "Atmospheric Chemistry and Physics". However, I have some concern about the novelty of the paper. The SSW events, their impacts, and predictions have been widely explored in literature, but this study shows little review on the previous studies (e.g., Charlton et al. 2007JC; Wang et al. 2019ACP; Rao et al. 2018JGR; Zhang et al. 2019Atmos...). Two SSWs were observed in February 2018 and January 2019, and the microwave

radiometer measurements also cover the February 2018 SSW. Therefore, I suggest to add some comparison between the two SSWs. What's the main difference in the ozone distribution between the two most recent SSWs? The authors also emphasize the relationship between the ozone and the SSW split. Because the February 2018 SSW is a typical polar vortex event, the analysis on this SSW is necessary to highlight the January 2019 SSW. In addition, the novelty of this study should be well stressed in the paper. The structure of this version can be further improved. Therefore, I recommend a major revision. Please see my specific comments below.

Specific comments: 1. The English language needs to be further improved, especially the simple tense. The authors used both the present and past tenses.

2. Page 1, Line 14: "number" change to "numbers"

3. Page 1, L18: Wired. Coriolis force is force (units: N), and radiation forcing (W/mˆ2) is energy. How are they balanced?

4. Page 1, L22: Add "one of" before "the most dramastic..."

5. Page 2, L1: Add the most recent and relevant references.

6. Page 2, L1: Ill sentence: "...lead to ... increase up to ..."

7. Page 2, L11-12: Ill sentence: "increases of up to ... and in the oder...."

8. Page 2, L13: "decrease of"??? Do you mean "decrease by"?

9. Page 2, L13: The 2018 SSW should be well reviewed to well compare the two SSW events.

10. Page 3, L15; Page 5, L12-13; Page 9, L11-12: The 2018 SSW event should be incorporated in this study, and you can extract the data. To further improve the quality of the paper, you can make full use of those data.

11. Page 4, L17: Change to "Hagen et al. (2018)". Change to "Figures 2 and 3"

12. Page 5, L31: Should be "first day"

13. Page 5, L22: Should be "the atmospheric component"

14. Page 6, L14-15: The two most recent SSW events are reported in Rao et al. (2018, 2019, 2020)

15. Page 6, L24: The SSW definition of WMO is based on the zonal wind at 60N, not from 60-90N. If your definition is different, please clarify.

16. Page 6, L25, L27: The reversal of wind is fast; how could it last for one month? Do you mean the easterlies last for one month?

17. Page 6, L26: Changed "reform" to "reappear". "Reform" does not mean "form again".

18. Page 7, L1-7: Add some references, because the process of the 2019 SSW has been reported recently in Rao et al. (2019, 2020)

19. Page 7, L9-11: Could you add some comparison between Observations at Ny-Ålesund and those in Wang et al. (2019ACP).

20. Page 7, L15: Do you mean that the elevated stratopause is only observed for split SSW. What did you observe for displacement SSWs?

21. Page 7, L16-18: Did Matthias et al (2012) and Limpasuvan et al. (2016) only composite the split SSWs, or all SSWs? Please clarify. It will confuse readers, because those sentence is following your statement for the split SSW. But the 2019 SSW is not a typical split SSW (Rao et al. 2019, 2020JGR). For the SSW type, also refer to Table 1 in Rao et al. (2019b, doi: 10.1029/2019JD030900), Charlton and Polvani (2007), and Butler et al. (2015).

22. Page 7, L29: The meridional wind in autumn is not that small. The result depends on the reference you compared to. If you compare u wind and v wind, it is indeed so.

23. Page 8, L6: What is "the rest of winter". Please be more specific.

24. Page 8, L7: If you smooth the time series, you can filter out the diurnal cycle. I suggest to remove the diurnal cycle and stress the variation related to the SSW.

25. Page 8, L10: I do not know how you obtain the peak magnitude and the period. You calculate them using the wavelet analysis? Or something else I missed?

26. Page 8, L12, L25, . . .: Change "inside of" to "inside"

27. Page 8, L14-15: Non-informative sentence. Even a nonsense. Please delete.

28. Page 8, L17: Contradict with those sentences if Matthias et al (2012) and Limpa-suvan et al. (2016) composite all SSWs (Page 7, L16-18).

29. Page 11, L1: Many reference show the important role of the planetary waves during SSWs. Please add more from the most recent publications.

30. Page 11, L16-19: Can you add some references?

31. Page 11, L24: I do not understand this sentence. Please clarify.

32. Page 11, L30: The equations should be put much earlier. You used the wave 1-2 in early part of the paper.

33. Page 12, L5-7: This sentence is too long, and seems ill.

34. Page 12, L12-13: Ill sentence.

35. Page 12, L29: The tense is change to the past. But you use the present tense most of the time.

36. Page 12, L33: What is the "period considered here"? Be specific. Add a "comma" after "therefore"

37. Page 13, L3: Replace "reform" with other verbs.

38. Page 13, L3-4: "was . . .. and . . . depends" The tense is rather confusing.

---

## Author Comment (AC1) · 4 Jun 2020

**Response to the Referees**

Franziska Schranz, Jonas Hagen, Gunter Stober, Klemens Hocke, Axel Murk,
Niklaus Kämpfer

May 27, 2020

Thank you for the comments and suggestions and also for pointing out small corrections and typos. The technical corrections have been implemented. Remarks and questions are addressed in a point-to-point response below.

In general we followed the suggestions of Referee 1 and highlight the analysis of the small-scale spatial ozone gradients. The discussion of the January 2019 SSW is meant to support the understanding of the temporal evolution of the ozone gradients at Ny-Ålesund. We therefore refrain from a more detailed discussion of the SSW which was suggested by Referee 2 in comments 9, 10 and 14.

**Response to Referee 1**

1)

P6, L23-25: I guess you mean that "between 60° and 90° N" refers to the reversal of the latitudinal temperature gradient at 10hPa but not to the reversal of the zonal mean zonal wind. Typically, it is the reversal of the zonal wind at 60° N what defines the occurrence of a major SSW, as it is indicated some lines above in the same page.

We use the definition of a major SSW by McInturff (1978) which is referred to as the WMO definition. It uses the reversal of the latitudinal temperature gradient and the reversal of the zonal mean zonal wind poleward of 60° latitude: "A stratospheric warming can be said to be major if at 10 mb or below the latitudinal mean temperature increases poleward from 60° latitude and an associated circulation reversal is observed (i.e., mean westerly winds poleward of 60° latitude are succeeded by mean easterlies in the same area)." We are aware that there are different definitions of major SSW conditions. In this clear case of a major SSW, the exact definition has no bearing on the final conclusions.

2)

P6, L30-31: Please describe more in detail the criterion for the identification of the polar vortex edge.

The identification of the polar vortex edge is now described as follows:

For a given pressure level the polar vortex is determined as the geopotential height (GPH) contour north of 15° N with the highest absolute wind speed compared to other GPH contours at the same pressure level. The GPH, and wind data are taken from ECMWF and the method is discussed in detail in Scheiben et al. (2012).

3)

P7, L1: It is indicated that vortex started to shift notably around 20 December, but the vortex structure is only shown from 25 December. I think it would be great to show its structure before 20 December to appreciate the shift.

The polar vortex contour is now additionally shown for 22 December 2018 as well as for 22 October 2018, 4 February 2019 and 3 March 2019.

4)

P7, L23-24: I am confused. I am not sure which level the authors are referring to, because in the stratosphere and most of the mesosphere, the winds remain westward the whole December except for some days in mid-December. Please rewrite this sentence.

The sentence has been changed.

Except for a few days in mid December, it stayed westward for the whole month in the stratosphere and lower mesosphere.

5)

P8, L12: Figure 7 only shows the water vapor from 1st October. However, since the effective descent rate of H2O is measured for the period of 15 September-1 November, I would start the figure on 15 September too.

All figures which show a time series have been modified and show the time period from 15 September 2018 – 1 April 2019.

6)

P11, L15: Figure 12 shows only the plots for a specific day of October, December and March but not the monthly field. I recommend the authors computing the monthly means of the fields if they want to show the monthly state.

In the monthly mean plots the characteristic state of the atmosphere which we want to discuss becomes blurry. We therefore decided to show the ozone and wind fields for more dates.

See Fig. 12.

7)

P11, L24-27: The wave activity is already low in January and the ozone gradient is then increasing by that time. However, by the end of January and the first fortnight of February the ozone gradient decreases and the wave activity is still low. How do you explain that?

The additional plots in Fig. 12 show that in the end of January and in the first fortnight of February the ozone was well mixed inside of the newly formed polar vortex.

In the beginning of February a weak polar vortex reestablished. Ozone is well mixed inside the newly formed vortex (Fig. 12i) and the gradient amplitude drops again.

8)

P12, L4: I think the wind is mainly southward in October.

We rewrote the sentence.

At the beginning of the winter season in October a stable polar vortex evolves with a strong zonal wind, which blocks the meridional transport of ozone from the mid-latitudes into the polar cap resulting in a 90° angle difference between the ozone gradient and the wind vector (e.g. towards the end of October the ozone gradient points southward and the wind vector points eastward).

9)

P12, L9-10: so, no specific effects of the SSW?

At 3 hPa the SSW first leads to varying gradient amplitudes when the vortex shifts and splits and ozone rich air from the midlatitudes enters the polar cap region and filamentary ozone structures develop. A low ozone pocket which passes at Ny-Ålesund leads to a drop in the gradient amplitude for a few days because the ozone distribution inside of the pocket is homogeneous. The following sentences were added.

During the SSW Figs. 12b–g) show the inflow of midlatitude air into the polar region and filamentary structures lead to variations in the gradient amplitude. In mid January a low ozone pocket crosses Ny-Ålesund (Fig. 12h) which shows very low gradient amplitudes.

10)

Figure 1: I cannot see the red square indicating the location of Ny-Ålesund.

The red square has been added.

**Response to Referee 2**

The numbers correspond to the comments of Referee 2.

3)

P1, L18: Wired. Coriolis force is force (units: N), and radiation forcing (W/m^2) is energy. How are they balanced?

This is true. It is the pressure gradient force which is balanced by the Coriolis force.

The polar vortex is a cyclonic wind system which forms in autumn from the balance between the Coriolis force and the pressure gradient force between the pole and the midlatitudes, which results form the radiative cooling of the polar middle atmosphere in the absence of solar heating.

5)

P2, L1: Add the most recent and relevant references.

The following references were added: Hocke et al. (2015) and Manney et al. (2009).

15)

P6, L24: The SSW definition of WMO is based on the zonal wind at 60° N, not from 60-90° N. If your definition is different, please clarify.

See point 1 in the response to Referee 1.

18)

P7, L1-7: Add some references, because the process of the 2019 SSW has been reported recently in Rao et al. (2019, 2020).

The references were added.

For altitudes below 10 hPa (about 30 km) the SSW was already discussed in Rao et al. (2019, 2020).

19)

P7, L9-11: Could you add some comparison between Observations at Ny-Ålesund and those in Wang et al. (2019).

Wang 2019 observed the SSW in February 2018 and not the SSW in Dec 2018/ Jan 2019.

20)

P7, L15: Do you mean that the elevated stratopause is only observed for split SSW. What did you observe for displacement SSWs?

According to Chandran et al. (2013) 68% of SSW with and elevated stratopause are split type events whereas the rest are displacement events. In this study we do not discuss other SSWs and focus on the Winter 2018/2019 event. The sentence was modified accordingly.

The elevated stratopause is a phenomenon which can occur after an SSW and has been previously observed (e.g. Manney et al., 2008). In a model climatology Chandran et al. (2013) found that 68 % of the SSWs with elevated stratopause are split type events and the remaining 32 % are displacement events.

21)

P7, L16-18: Did Matthias et al (2012) and Limpasuvan et al. (2016) only composite the split SSWs, or all SSWs? Please clarify. It will confuse readers, because those sentence is following your statement for the split SSW. But the 2019 SSW is not a typical split SSW (Rao et al. 2019, 2020JGR). For the SSW type, also refer to Table 1 in Rao et al. (2019b, doi: 10.1029/2019JD030900), Charlton and Polvani (2007), and Butler et al. (2015).

Matthias et al. (2012) and Limpasuvan et al. (2016) consider major SSWs with an elevated stratopause which contains both split and displacement type events.

Matthias et al. (2012) compiled a composite of SSW events with an elevated stratopause between 1998 to 2011 using ECMWF data and MF-radar observations revealing the formation of an elevated stratopause in the mesospheric wind as strong zonal wind enhancement after the SSW. Later, Limpasuvan et al. (2016) made a composite analysis of SSWs with an elevated stratopause using WACCM data and explained the occurrence of an elevated stratopause with strong, wave induced downwelling in the mesosphere which leads to adiabatic warming.

24)

P8, L7: If you smooth the time series, you can filter out the diurnal cycle. I suggest to remove the diurnal cycle and stress the variation related to the SSW.

The SSW took place in the beginning of January. At a latitude of 79° there is polar night during this period even in the mesosphere and hence no sun induced diurnal ozone cycle which could mask the effect of the SSW. Schranz et al. (2018) provides a discussion of the diurnal ozone cycle in the stratosphere and mesosphere at the location of Ny-Ålesund.

25)

P8, L10: I do not know how you obtain the peak magnitude and the period. You calculate them using the wavelet analysis? Or something else I missed?

Yes it is a wavelet-like approach. The time series were filtered with a digital non-recursive, zero-phase finite-impulse response filter using a Hamming window whose length is 3 times the centre period (Hocke and Kämpfer, 2008; Hocke, 2009).

Using a wavelet-like approach which is described in Hocke and Kämpfer (2008); Hocke (2009) we find peak-to-peak amplitudes of 0.8 ppm and periods of 3–4 days in November.

28)

P8, L17: Contradict with those sentences if Matthias et al. (2012) and Limpasuvan et al. (2016) composite all SSWs (Page 7, L16-18).

Matthias et al. (2012) and Limpasuvan et al. (2016) use SSW events with an elevated stratopause and find enhanced zonal wind and downwelling which is exactly what we see.

29)

P11, L1: Many reference show the important role of the planetary waves during SSWs. Please add more from the most recent publications.

The following sentence was added.

Large amplification of the planetary wave amplitude was observed prior to and during SSWs (e.g. Lawrence and Manney, 2020; Matthias and Ern, 2018).

30)

P11, L16-19: Can you add some references?

The following references were added: Meek et al. (2017) and Manney et al. (1994).

In October the amplitudes of the planetary waves 1 and 2 are low and the polar vortex is mainly centred at the pole and the zonal wind field is zonally symmetric which leads to a transport barrier between midlatitude and polar air (e.g. Meek et al., 2017; Manney et al., 1994).

31)

P11, L24: I do not understand this sentence. Please clarify.

Inside of the polar vortex ozone is not symmetrically distributed but consists of filaments with lower and higher ozone VMR which show strong horizontal gradients. The filaments pass at Ny-Ålesund when the polar vortex rotates. The sentence was reformulated.

During the SSW Figs. 12b–g) show the inflow of midlatitude air into the polar region and filamentary structures in the ozone field lead to variations in the gradient amplitude.

32)

P11, L30: The equations should be put much earlier. You used the wave 1-2 in early part of the paper.

This equation shows only the reconstruction of the wind field from wave 1 and 2 and not the algorithm for extracting wave amplitude and phase. This is described in detail in Baumgarten and Stober (2019) and we refer to this source earlier in the manuscript.

**References**

[revised manuscript text omitted]